# Effects of Fertilizer Application Patterns on Foxtail Millet Root Morphological Construction and Yield Formation during the Reproductive Stage in the Loess Plateau of China

Tianyou Zhou [1], Huaping Zhang [1], Qinhui Liu [1], Lichao Wei [1] and Xiaolin Wang [1,2,*]

[1] College of Life Sciences, Yulin University, Yulin 719000, China; 18302491798@163.com (T.Z.); zhanghp820718@163.com (H.Z.); liuqinhui574@163.com (Q.L.); wlc18065176991@163.com (L.W.)

[2] Engineering and Technology Research Center of Water Saving for Crops in Arid Area of Northern Shaanxi, Yulin 719000, China

[*] Correspondence: wangxl8304@163.com; Tel.: +86-151-2997-8707

**Abstract:** With crop yields continually increasing, chemical fertilizer consumption in China is increasing in parallel. The excessive use of synthetic fertilizer can lead to soil compaction, acidification, and degeneration, which can all be mitigated through additional organic manure application. The combined application of organic manure and inorganic fertilizer plays crucial roles in the root morphology and yield formation of dryland crops. In this study, foxtail millet (Chang Sheng 07) was used as the experimental material and sown in a dry farming area with five different fertilizing patterns, which were composed of chemical N, P, organic manure, and microbial manure. These patterns included a single application of 45 kg·ha$^{-1}$ of nitrogen fertilizer as the control ($N_{45}$), the combined application of 60 kg·ha$^{-1}$ of nitrogen fertilizer with 30 kg·ha$^{-1}$ of phosphorus fertilizer ($N_{60}P_{30}$), the combined application of 90 kg·ha$^{-1}$ of nitrogen fertilizer with 45 kg·ha$^{-1}$ of phosphorus fertilizer ($N_{90}P_{45}$), 60 kg·ha$^{-1}$ of nitrogen fertilizer and 40 kg·ha$^{-1}$ of phosphorus fertilizer with 2000 kg·ha$^{-1}$ of organic manure ($N_{60}P_{40}$-O), and 60 kg·ha$^{-1}$ of nitrogen fertilizer and 40 kg·ha$^{-1}$ of phosphorus fertilizer with 5 kg·ha$^{-1}$ of microbial manure ($N_{60}P_{40}$-M). Each treatment was performed with four repeats. The results show that (1) the different fertilization patterns had significant effects on the morphological construction of foxtail millet roots, and the root length (*RL*) with $N_{60}P_{40}$–O underwent a significant increase of 88.23% and 61.59% in the two stages, respectively, (2) as confirmed by the correlation analysis, the root surface area (*RSA*) was positively correlated with the *RL* and root volume (*RV*), (3) the yields with $N_{90}P_{45}$ and $N_{60}P_{30}$ exhibited a significant increase of 54.43% and 59.86%, and those with $N_{60}P_{40}$–O and $N_{60}P_{40}$–M stably increased by 13.12–24.11% compared to those with $N_{45}$, and (4) the water use efficiency (*WUE*) of foxtail millet under the $N_{60}P_{30}$ and $N_{90}P_{45}$ patterns significantly increased by 33.40–62.39%, while that under the $N_{60}P_{40}$–O and $N_{60}P_{40}$–M patterns increased by 12.89–29.20%. In summary, the application of additional organic matter and microbial manure promoted the morphological construction of foxtail millet roots, led to better stability in grain production, and is an ecofriendly option in terms of sustainable land use.

**Keywords:** dryland foxtail millet; root morphology; fertilization model; yield; water use efficiency





## 1. Introduction

In China, the annual consumption of chemical fertilizer accounts for 35% of the total global consumption [1]. In particular, inorganic fertilizers such as urea, which contains 46% nitrogen, pose a significant threat to human health through their excessive application [2]. Excessive synthetic fertilizer application disturbs the soil nutrient balance and soil physical properties, reducing crop yield and fertilizer use efficiency [3,4] and severely affecting the agricultural and ecological environment [5,6]. Therefore, our continued overreliance on chemical fertilizers for crop production is not sustainable. There is growing interest in alternative synthetic fertilizers with organic manure for advanced farming as a means to

decrease the problems associated with the use of these products without compromising crop productivity. Organic manure can optimize soil structure and nutrient content, maintain a relatively stable level of soil organic matter, improve soil quality, and enhance farmland stress resistance [7]. Soil microbial communities play a crucial role in maintaining soil fertility at the appropriate levels and in improving soil structure by affecting the aggregation of soil particles [8], while microbial fertilizer can improve the relationship between plant roots and soil water, provide drought resistance protection, and simultaneously render plants less susceptible to certain soil-borne diseases, including diseases caused by fungi that produce additional fungal toxins, and they can also reduce the occurrence of pests [9]. Microbial fertilizers are applied to seeds or farmland with the aim to increase soil fertility and plant growth by enhancing the diversity and biological activity of microorganisms in the rhizosphere. Soil is a complex system influenced by multiple factors [10] and improving the beneficial microbial community in the soil is important for the biogeochemical cycling of inorganic and organic nutrients, especially in the rhizosphere, where microorganism activity increases the availability of plant nutrients, soil quality, and crop root growth. However, the relatively low nutrient content of organic manure makes it unsuitable for crop slow-release purposes. Through the combination of organic plant nutrients, organic manure can be afforded the ability of inorganic fertilizers to promote crop growth in isolated applications [11], and this method has been gaining attention in recent years. Increasing the amount of organic manure and decreasing the amount of chemical fertilizers used are important for augmenting the soil organic matter content [12,13].

In a recent study with a long-term single application of chemical fertilizers, the nitrogen absorption of corn showed a trend of initially increasing and then decreasing [14]. The absorption of water and mineral nutrients in soil mainly relies on the crop root system [15]. Root morphological traits and spatial distribution play important roles in the growth and development of plants. Root system morphological traits include root length, root surface area, volume and density at various depths, and root length in different diameter classes [14,16]. Nitrogen fertilizer can significantly improve the root activity, root length, root surface, root biomass, and root volume of foxtail millet, and an appropriate amount of nitrogen fertilizer can improve its root activity, while excessive nitrogen application inhibits it [17]. Wei et al. showed that the combination of organic and inorganic fertilizers can significantly improve root length, surface area, and volume, enhance rice root vitality as well as nutrient absorption and utilization by roots, and provide sufficient nutrients for plant growth, which is beneficial for the accumulation of dry matter in various parts of the plant and to increase rice yield [18]. Hence, the combined application of organic and inorganic fertilizers or the application of organic fertilizers alone could improve nitrogen fertilizer utilization efficiency and be conducive to the robust development of roots, as well as yield increase [19,20].

Under different nitrogen application treatments, the amount of nitrogen fertilizer applied has been found to be closely related to the foxtail millet yield and *WUE*, and the impacts of different water–manure coupling treatments on yield varied greatly [21]. Under different fertilizer treatments, a significant impact on crop yield has been determined with the application of a phosphorus and potassium fertilizer [22]. Using the proper input and proportion of nitrogen, phosphorus, and potassium fertilizer can increase crop growth, enhance *WUE*, and improve soil fertility [23]. For this reason, the basic raw material for, and primary factor influencing, soil fertility is the use of an ordinary chemical fertilizer. If an ordinary chemical fertilizer is applied alone for an extended period, the soil will harden, which is not conducive to crop growth. Applying a reasonable proportion of organic and inorganic fertilizer could promote the morphological growth of foxtail millet roots and, correspondingly, increase yield [13]. In summary, the effect of fertilizer application is related to the morphological development of crop roots and improves crop yield.

Foxtail millet (*Setaria italica* (L.) *Beauv.*) is representative of the origins of dry farming agriculture in northern China and is one of the oldest cultivated crops, with a history of approximately 8000 years. It is mainly distributed in arid and semi-arid regions in the north of China [24]. It has the characteristics of tolerance to drought and high adaptability, and its grains are rich in nutrients and are called millet after shelling [19]. It is an important economic crop in the arid farming areas of the Loess Plateau. However, the restriction of its natural environment has occurred due to factors such as unexpected high temperatures, heavy rainfall, and so on [25], and increases in productivity levels, the use of backward tillage technology, and the use of chemical fertilizers have resulted in the soil ecological system constant degeneration [24,25]. Furthermore, this chain reaction has not only caused agricultural costs to increase, but it has also led to a waste of fertilizer energy, a significant decrease in the fertilizer utilization rate, the useless leaching of soil and environmental pollution, as well as soil acidification and the imbalance of microbial communities in most of the northern farmlands [26]. In the semi-arid region of northern China, the severe response of local crops to soil degeneration has caused difficulties in improving the yield of foxtail millet in recent years [26]. The yield of foxtail millet is closely related to root morphology in dryland cultivation, being determined not only by the connecting organ between the aboveground and underground part of the millet plant [9] but also by the key root physiological function that influences the capacity of the crop to absorb water and nutrients [27]. The rational application of fertilizer is an essential condition for the optimization of root morphology and increase in the yield of millet [9,28]. By adjusting the amount of fertilizer and using the appropriate ratio of manure, sustainable crop production can be achieved [29,30].

In this study, we used the foxtail millet cultivar "Changsheng 07", which is characterized by significant resistance and a morphological structure suitable for a high grain yield. It was assumed in the current research that organic manure, microbial manure, and chemical fertilizer, in a combined application, could regulate and reconstitute the foxtail millet root system and increase the final yield and *WUE* in a dryland farm region. The specific objectives of the present research were, firstly, to determine the most effective strategy and the economic effects of organic and inorganic fertilizer application on the crop growth and yield, and, secondly, to provide a theoretical basis for the rational combined application of ecological fertilizers to improve dryland crop yield and resource use efficiency.

## 2. Materials and Methods

### 2.1. Field Experimental Sites

The field experiments were conducted from April to October in 2017 and 2018 in Hengshan District, Yulin City, Shaanxi Province (Figure 1), with crisscrossing gullies and deep soil layers. The climate is temperate arid and semi-arid, with rainfed agricultural areas. The altitude is 1232 m, with a mean annual temperature, rainfall, evaporation, and sunshine duration of 9.6 °C, 444 mm, 1211 mm, and 2644 h, respectively (Figure 2). The frost-free period lasts approximately 146 days. The rainfall is concentrated from July to September every year, accounting for more than 60% of the total rainfall. During the foxtail millet growth period in this study, the rainfall and the annual sunshine duration were 513.2 mm and 2458 h, respectively. The soil is loessal soil, with organic matter, total nitrogen, alkali hydrolyzed nitrogen, available phosphorus, and available potassium contents of 3.2 g·kg$^{-1}$, 0.3 g·kg$^{-1}$, 18.9 mg·kg$^{-1}$, 6.2 mg·kg$^{-1}$, and 66.0 mg·kg$^{-1}$, respectively.

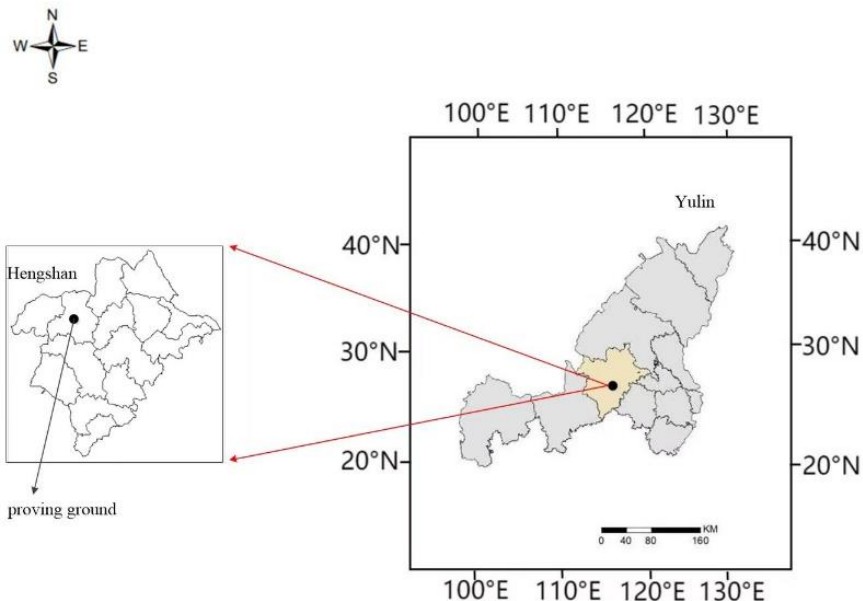

**Figure 1.** Experimental site location.

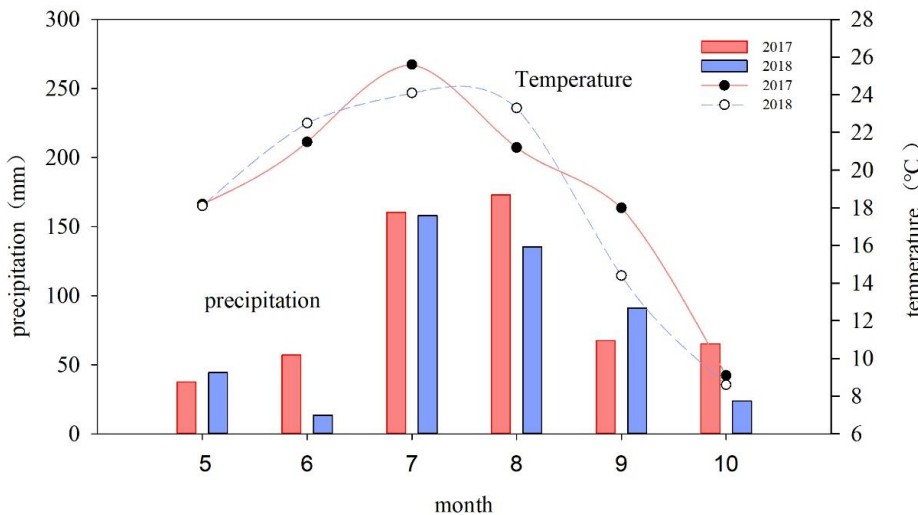

**Figure 2.** Changes in average temperature and precipitation in Hengshan District during 2017–2018.

*2.2. Materials*

The selected foxtail millet cultivar was "Changsheng 07", which was bred at the Millet Research Institute, Shanxi Academy of Agricultural Sciences, with spindle spikes, a moderate plant height, a high, stable yield, and resistance to lodge and disease, seeing as this plant is also a commodity. The foxtail millet was suitable for a frost-free period of more than 150 days in dryland and planting in spring. The seeds in the experiments were selected based on the appearance of full grains, with the following considered desirable: a uniform size, color, and lack of damage.

The nitrogen fertilizer contained N at a level over 46.4% and was produced by the Inner Mongolia Boyuan International Agricultural Means of Production Co., Ltd., Ordos City, China. The calcium superphosphate, with a content of 12% $P_2O_5$, was manufactured by Hanzhong Tangfeng Chemical Co., Ltd., Hanzhong, China. The organic manure (33% organic matter) was produced by Huai'an Huisheng Agriculture, Forestry and Horticulture Development Co., Ltd., Huai'an, China, with its NPK content greater than or equal to 3%. The microbial manure was produced by Shandong Lvlong Biotechnology Co., Ltd., Zibo, China, with an effective viable bacteria content over 20 billion·$g^{-1}$, obtained as a

wettable powder composed of a variety of high-activity probiotics and their metabolites, which can fix nitrogen, solubilize phosphate, and release potassium.

### 2.3. Experimental Design and Management

From 2017 to 2018, at the end of April, regular cultivation was employed for land preparation, plowing, film covering, ridging, and plot division. The subplots were designed to be 4 m wide × 5 m long, with a plot area of 20 m$^2$ each. The distance between each plot was 0.5 m to prevent the soil water and nutrients from leaching. After wind-drying the seeds, bunch planting on the 3–5 cm film side, soil covering, and compaction were carried out.

The experiments were replicated 4 times in a random block design, adopting five fertilization patterns that included a single application of nitrogen fertilizer as a control ($N_{45}$), the combined application of nitrogen fertilizer with phosphorus fertilizer ($N_{60}P_{30}$ and $N_{90}P_{45}$ had different fertilizer amounts), nitrogen fertilizer with organic manure ($N_{60}P_{40}$–O), and nitrogen fertilizer with phosphorus fertilizer and microbial manure ($N_{60}P_{40}$–M). The total manure application amounts are shown in Table 1.

**Table 1.** The experimental designs of different fertilizers (kg·ha$^{-1}$).

| Treatment | N | P$_2$O$_5$ | Organic Manure * | Microbial Manure * |
|---|---|---|---|---|
| $N_{45}$ | 45 | 0 | 0 | 0 |
| $N_{60}P_{30}$ | 60 | 30 | 0 | 0 |
| $N_{90}P_{45}$ | 90 | 45 | 0 | 0 |
| $N_{60}P_{40}$–O | 60 | 40 | 2000 | 0 |
| $N_{60}P_{40}$–M | 60 | 40 | 0 | 5 |

* Note: for organic manure (33% organic matter), the NPK content was greater than or equal to 3%. For microbial manure, the effective viable bacteria content was over 20 billion·g$^{-1}$, obtained as a wettable powder composed of a variety of high-activity probiotics and their metabolites, which can fix nitrogen, solubilize phosphate, and release potassium.

The planting density of the foxtail millet was 180,000 plants·ha$^{-1}$. The seedlings were established based on their density. The fertilizer application time was the same for each treatment, and, to ensure the timely irrigation of crops due to water shortages, two inter-tillage periods, artificial weeding, and other management measures were used, according to the requirements for field experiments. Finally, the foxtail millet was harvested uniformly when it was mature.

### 2.4. Determination Index and Method

2.4.1. Determination of Root Morphology of Foxtail Millet

The roots were randomly excavated, with three samples reflecting the plant relatively similar growth for each treatment subplot, with a total of 45 plants in the pustulation and maturity stages, respectively. A drilling method of the soil profile root was adopted to dig out soil samples 50 cm long, 40 cm wide across the row, and 40 cm deep. Soil samples of 10 cm × 10 cm × 10 cm were collected, avoiding the root parts. We used a nylon mesh sieve to remove most of the soil and then washed the samples under a slow flow of water, removing the residual soil with a brush. Finally, the fine roots were collected in a recording bag and placed in a zip lock bag with a serial number and stored at 4 °C in a fridge for testing.

Prior to measurement, the sample was cleaned with distilled water and placed on a transparent resin plastic sheet without overlapping or crossing. Then, a root scanner (Epson V700, Beijing, China) was used to scan all the roots in the pustulation stage and maturity stage. Then, the scanned root images were analyzed using the Win RHIZO Program 2007, including the root length (*RL*), root surface area (*RSA*), root volume (*RV*), root biomass (*RB*), root diameter (*RD*), and other morphological indexes. The scanned roots were deoxidized at 105 °C for 30 min in the oven. After that, the root biomass was measured through drying to a constant weight at 75–85 °C.

2.4.2. Determination of the Yield and Composition of Foxtail Millet

In total, 20 same-plant samples were randomly collected from the middle of each plot using the soil drilling method. We collected data on the profile root at maturity and after harvesting, the spike lengths, spike diameter, weight per spike, grain weight per spike, 1000-grain weight, and yields, as measured in the laboratory. The spike length (cm) was measured from the top of the spike to the base using a tape, and the spike diameter (mm) was measured with Vernier calipers, with each measurement taken at the center of the spike. The foxtail millet spike was cut at the base and weighted with an electronic balance with an accuracy of 0.001 to record the data. Then, all the mature spikes were harvested. After the foxtail millet (including 20 plants) was naturally dried, its weight per spike (g), 1000-millet weight (g), and total dry weight (g) were determined for each subplot with an electronic balance precision of 0.001, and the total yield of each subplot was calculated.

2.4.3. *WUE* Calculation

In the early sowing and harvest stages, soil samples were collected at 10 cm intervals over a depth of 0~100 cm. After collection, the samples were immediately sealed within plastic bags and transported to the laboratory. The soil moisture content was determined using the oven drying method. The water consumption was calculated, and the evapotranspiration (ET, mm) of each subplot was calculated using Formula (1):

$$ET = W1 - W2 + P \tag{1}$$

where P is the effective precipitation during the growth period of foxtail millet, W1 is the early sowing period, and W2 is the harvest period.

The *WUE* was calculated based on Equation (2):

$$WUE = \frac{Y(\text{kg·hm})}{\text{ET(mm)}} \tag{2}$$

where *WUE* = water use efficiency (kg mm$^{-1}$ hm$^{-1}$), ET = evapotranspiration (mm), and $Y$ = yield (kg hm$^{-1}$).

*2.5. Data Analysis and Statistics*

SPSS Statistics 27.0 was used to assess the main effects of different fertilizer application patterns on the foxtail millet root, yield, and *WUE*. We then conducted a one-way analysis of variance, multiple comparison of the LSD, and correlation analysis at $\alpha = 0.05$, and the data in the chart are the mean $\pm$ standard deviation. The figures were created using Origin 2017, Sigma subplot 12.5, and ArcGIS 10.8.

**3. Results**

*3.1. Effects of Different Fertilizers on the Total Root Length (RL) of Foxtail Millet*

The *RL* of foxtail millet showed differences under different fertilizer application patterns in the pustulation and maturity stages (Figure 3). Compared with N$_{45}$, the *RL* increased by 44.28% at N$_{60}$P$_{30}$ and 88.23% at N$_{60}$P$_{40}$–O in the pustulation stage ($p < 0.05$). Furthermore, the *RL* values of N$_{60}$P$_{30}$, N$_{90}$P$_{45}$, and N$_{60}$P$_{40}$–O increased by 78.03%, 61.63% and 61.59% ($p < 0.05$), respectively, but only increased by 0.57% in the case of N$_{60}$P$_{40}$–M, indicating a non-significant level ($p > 0.05$), in the maturity stage. Nevertheless, the *RL* of N$_{90}$P$_{45}$ decreased compared with N$_{60}$P$_{30}$, indicating that the increase in the NP application rate could not significantly increase the *RL* but had a better regulation effect on the root growth. Under N$_{60}$P$_{40}$–O, the *RL* of the foxtail millet improved, indicating that the combined application of organic manure with a reasonable amount of nitrogen and phosphorus had a promoting effect.

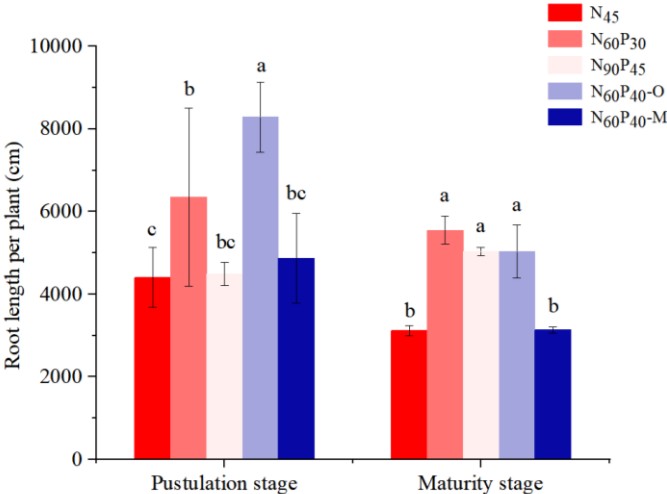

**Figure 3.** Effects of different fertilizer application patterns on the root length of millet. Note: $N_{45}$ (control); $N_{60}P_{30}$, $N_{90}P_{45}$ (different application amounts); $N_{60}P_{40}$–O (nitrogen and phosphorus fertilizer combined with organic manure); $N_{60}P_{40}$–M (nitrogen and phosphorus fertilizer combined with microbial manure). Different lowercase letters in the upper part of the bar chart indicate significant differences between treatments at $\alpha$ = 0.05. Bars indicate SD.

### 3.2. Effects of Different Fertilizer Combinations on the Root Surface Area (RSA) of Foxtail Millet

The root surface area was reconfigured and adjusted during the pustulation stage (Figure 4) according to the variation in the *RSA* for $N_{60}P_{30}$, $N_{60}P_{40}$–O, and $N_{60}P_{40}$–M, all of which significantly increased ($p < 0.05$) by 56.74%, 66.19%, and 20.64%, respectively, compared with $N_{45}$. Compared with $N_{45}$, the *RSA* for $N_{90}P_{45}$ decreased by 1.71% at a non-significant level ($p > 0.05$). In maturity, the *RSAs* of $N_{60}P_{30}$ and $N_{60}P_{40}$–O significantly increased by 24.69% and 16.78%, respectively, compared with $N_{45}$, and the *RSA* of $N_{60}P_{40}$–M decreased by 17.15% ($p < 0.05$), while that of $N_{90}P_{45}$ was not significantly different from that of $N_{45}$ ($p > 0.05$), increased by only 9.88%.

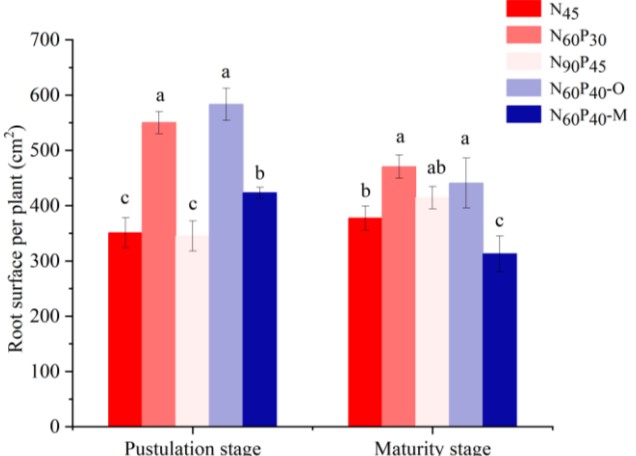

**Figure 4.** Effects of different fertilizer application patterns on the root area of millet. Note: $N_{45}$ (control); $N_{60}P_{30}$, $N_{90}P_{45}$ (different application amounts); $N_{60}P_{40}$–O (nitrogen and phosphorus fertilizer combined with organic manure); $N_{60}P_{40}$–M (nitrogen and phosphorus fertilizer combined with microbial manure). Different lowercase letters in the upper part of the bar chart indicate significant differences between treatments at $\alpha$ = 0.05. Bars indicate SD.

### 3.3. Root Volume (RV) Change of Foxtail Millet in Five Fertilizer Patterns

There was a major effect of the combined application of different fertilizers on the foxtail millet *RV* (Figure 5). The *RV* differences between fertilizer treatments showed a

similar trend for both stages. In the pustulation stage, the *RV*s of $N_{60}P_{30}$ and $N_{60}P_{40}$–O significantly increased by 33.46% and 38.78% compared with that of $N_{45}$, and the *RV*s increased by 4.56% for $N_{90}P_{45}$ and 27.76% for $N_{60}P_{40}$–M. There was no significant difference in the *RV* among the $N_{60}P_{40}$–M, $N_{45}$, $N_{60}P_{30}$, $N_{90}P_{45}$, and $N_{60}P_{40}$–O patterns ($p > 0.05$). The *RV*s of $N_{60}P_{30}$, $N_{90}P_{45}$, and $N_{60}P_{40}$–M were not significantly different from the *RV* of $N_{45}$. In the maturity stage ($p > 0.05$), the *RV* of $N_{90}P_{45}$ decreased by 7.30%, while the *RV*s of $N_{60}P_{30}$ and $N_{60}P_{40}$–M increased by 5.91% and 10.95%, respectively. However, compared with $N_{45}$, the *RV* of $N_{60}P_{40}$–O increased by 18.64% ($p < 0.05$).

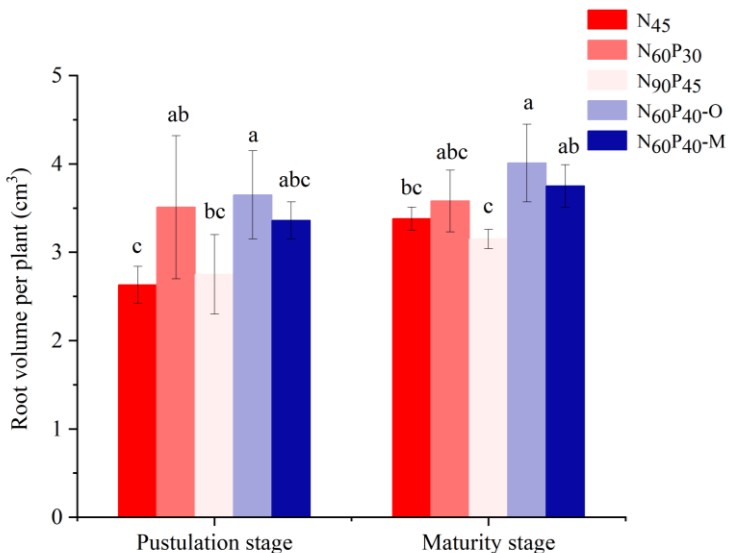

**Figure 5.** Root volume (*RV*) change of foxtail millet in five fertilizer patterns. Note: $N_{45}$ (control); $N_{60}P_{30}$, $N_{90}P_{45}$ (different application amounts); $N_{60}P_{40}$–O (nitrogen and phosphorus fertilizer combined with organic manure); $N_{60}P_{40}$–M (nitrogen and phosphorus fertilizer combined with microbial manure). Different lowercase letters in the upper part of the bar chart indicate significant differences between treatments at $\alpha = 0.05$. Bars indicate SD.

*3.4. Root Diameter (RD) Change of Foxtail Millet in Five Fertilizer Patterns*

The combined application of different fertilizers had a great impact on the foxtail millet *RD*, though to different extents (Figure 6). Compared with $N_{45}$, the *RD*s of $N_{90}P_{45}$, $N_{60}P_{40}$–O, and $N_{60}P_{40}$–M significantly ($p < 0.05$) increased by 35.12%, 87.63%, and 68.23%, respectively, in the pustulation stage, while there was no significant difference between the values for $N_{60}P_{30}$ and $N_{45}$ ($p > 0.05$). The *RD*s for the $N_{60}P_{30}$, $N_{90}P_{45}$, and $N_{60}P_{40}$–O patterns increased by 26.21%, 20.58%, and 33.40% compared with $N_{45}$ in the maturity stage, and the *RD* of $N_{60}P_{40}$–M decreased by 1.55%. There was no significant difference between the *RD*s of $N_{60}P_{30}$ and $N_{90}P_{45}$ ($p > 0.05$).

*3.5. Effects of Different Fertilizer Combinations on the Root Biomass (RB) of Foxtail Millet*

Root dry weight reflects root function and material accumulation ability, and fertilization mode showed a significant regulation effect on *RB*. In the pustulation stage, the *RB* for $N_{60}P_{30}$, $N_{90}P_{45}$, $N_{60}P_{40}$–O, and $N_{60}P_{40}$–M increased by 68.25%, 33.22%, 53.08%, and 52.60%, respectively ($p < 0.05$, Figure 7). Among them, the *RB* for $N_{60}P_{30}$ exhibited a significant increase compared to the other four patterns. There was no significant difference between the *RB*s for $N_{60}P_{40}$–O and $N_{60}P_{40}$–M ($p > 0.05$). In maturity, the *RB* of $N_{60}P_{30}$, $N_{60}P_{40}$–O, and $N_{60}P_{40}$–M increased significantly, and the *RB* of $N_{60}P_{40}$–O increased by 35.21% compared with $N_{45}$. The *RB*s of the $N_{60}P_{30}$ and $N_{60}P_{40}$–M patterns decreased by 19.01% and 14.08%, respectively, and that of $N_{90}P_{45}$ was not significantly different from that of $N_{45}$.

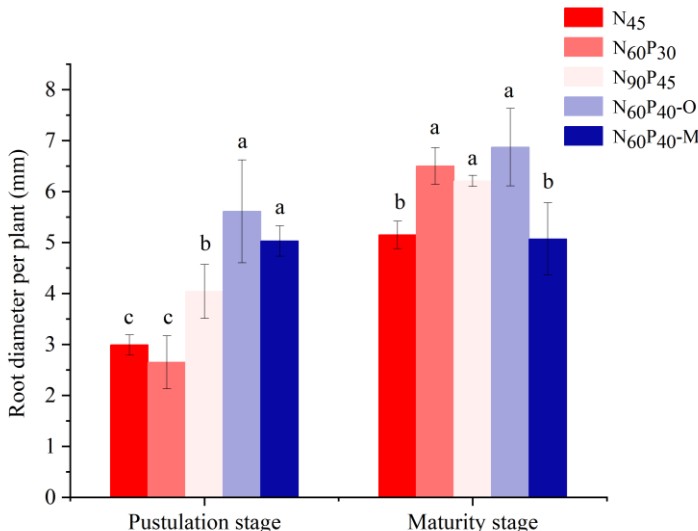

**Figure 6.** Effects of different fertilizer application patterns on the root diameter of millet. Note: $N_{45}$ (control); $N_{60}P_{30}$, $N_{90}P_{45}$ (different application amounts); $N_{60}P_{40}$–O (nitrogen and phosphorus fertilizer combined with organic manure); $N_{60}P_{40}$–M (nitrogen and phosphorus fertilizer combined with microbial manure). Different lowercase letters in the upper part of the bar chart indicate significant differences between treatments at $\alpha = 0.05$. Bars indicate SD.

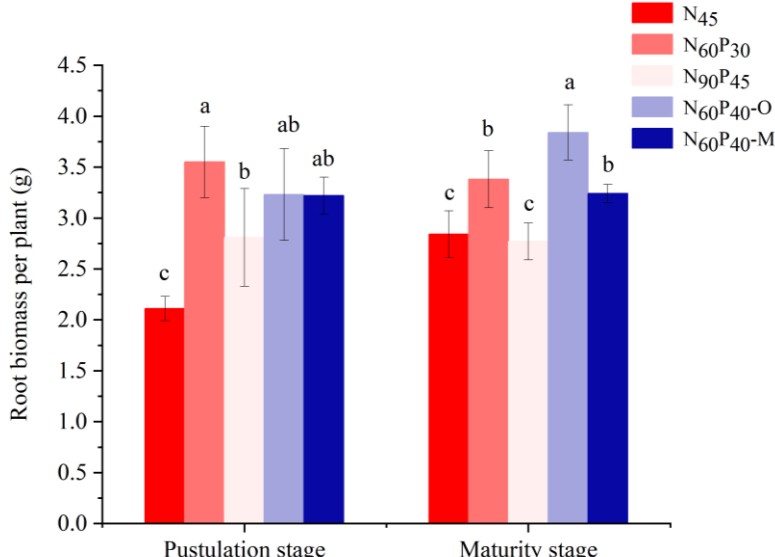

**Figure 7.** Effects of different fertilizer application patterns on the root biomass of millet. Note: $N_{45}$ (control); $N_{60}P_{30}$, $N_{90}P_{45}$ (different application amounts); $N_{60}P_{40}$–O (nitrogen and phosphorus fertilizer combined with organic manure); $N_{60}P_{40}$–M (nitrogen and phosphorus fertilizer combined with microbial manure). Different lowercase letters in the upper part of the bar chart indicate significant differences between treatments at $\alpha = 0.05$. Bars indicate SD.

*3.6. Correlation between Root Indexes and the Yield of Foxtail Millet*

There was a clear relation between the root indexes and yield during the foxtail millet pustulation stage in 2017–2018 (Figure 8). In 2017, there were highly significant correlations between the *RL* and *RSA* (0.82), the *RL* and *RV* (0.65), the *RSA* and *RD* (0.66), the *RSA* and *RV* (0.97), and the *RD* and *RV* (0.81) ($p < 0.01$). In 2018, there were positive correlations between the *RL* and *RSA* (0.78), the *RSA* and *RV* (0.69), and the *RV* and *RB* (0.79) ($p < 0.01$). The *RL*, *RSA*, and *RV* showed a stable positive correlation under the different fertilizer management strategies.

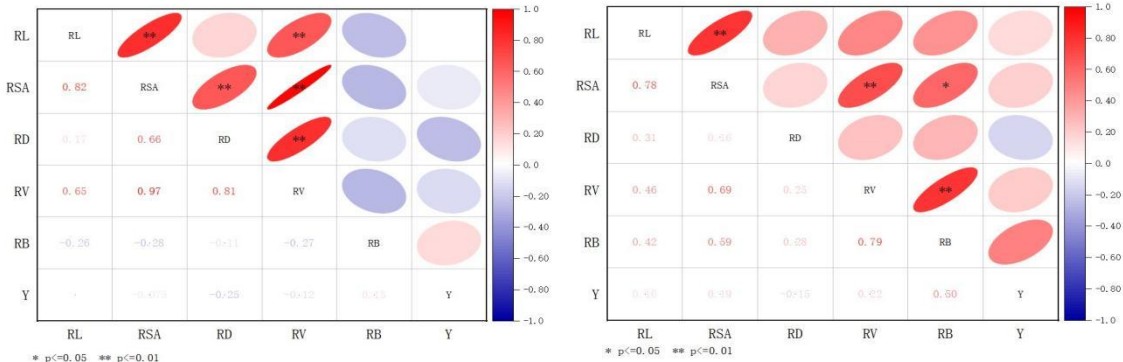

**Figure 8.** Correlation analysis between root index and yield in the pustulation period. Note: ** shows a significant correlation at the 0.01 level; * shows a significant correlation at the level of 0.05.

### 3.7. Effects of Different Fertilizer Application Patterns on Yield Composition

Combined organic and inorganic fertilizer application had a significant effect on the final grain yield (Table 2). The correlated factors that determined the yield were the spike diameter, weight per spike, and grain weight per spike over the two experimental years. The spike length for all other treatments was higher than that for $N_{45}$ by 2.92%, 11.54%, 4.23%, and 14.81%, respectively. Compared with $N_{45}$, the spike diameters of $N_{60}P_{30}$ and $N_{60}P_{40}$–O were lower by 10.99% and 3.66%; however, those of $N_{90}P_{45}$ and $N_{60}P_{40}$–M increased by 3.66% and 12.45%. The weight per spike of $N_{45}$ was significantly different from that of $N_{60}P_{30}$ and $N_{90}P_{45}$ ($p < 0.05$), increased significantly by 57.36% and 52.69%, respectively, while that of $N_{60}P_{40}$–O and $N_{60}P_{40}$–M increased by 22.94% and 14.42%. The grain weight per spike for $N_{60}P_{30}$, $N_{90}P_{45}$, $N_{60}P_{40}$–O, and $N_{60}P_{40}$–M was higher than that of $N_{45}$, increased by 59.91%, 53.92%, 23.84%, and 13.00%, respectively. There were significant differences in weight per spike between $N_{60}P_{30}$, $N_{90}P_{45}$, and $N_{45}$ ($p < 0.05$). There was no significant difference in 1000-grain weight among $N_{45}$, $N_{60}P_{30}$, $N_{90}P_{45}$, $N_{60}P_{40}$–O, and $N_{60}P_{40}$–M.

**Table 2.** Effects of different fertilizer application patterns on yield composition.

| Treatment | Spike Length/ (cm) | Spike Diameter/ (cm) | Weight per Spike/(g) | Grain Weight per Spike/(g) | 1000-Grain Weight/(g) |
|---|---|---|---|---|---|
| $N_{45}$ | 20.5 ± 3.2 a | 2.7 ± 0.2 ab | 19.5 ± 3.6 b | 15.7 ± 3.0 b | 3.0 ± 0.1 a |
| $N_{60}P_{30}$ | 21.1 ± 1.7 a | 2.4 ± 0.5 b | 30.7 ± 2.4 a | 25.1 ± 2.2 a | 3.1 ± 0.2 a |
| $N_{90}P_{45}$ | 22.9 ± 1.6 a | 2.8 ± 0.1 ab | 29.8 ± 5.5 a | 24.2 ± 4.7 a | 3.2 ± 0.1 a |
| $N_{60}P_{40}$–O | 21.4 ± 2.1 a | 2.6 ± 0.3 ab | 24.0 ± 2.6 ab | 19.4 ± 2.5 ab | 3.1 ± 0.1 a |
| $N_{60}P_{40}$–M | 23.6 ± 0.9 a | 3.1 ± 0.3 a | 22.3 ± 7.0 ab | 17.7 ± 5.6 ab | 3.1 ± 0.2 a |

Note: Different lowercase letters in the same column indicate significant differences at the $\alpha$ = 0.05 level.

### 3.8. Effects of Different Fertilizer Application Patterns on the Yield and WUE of Foxtail Millet

There were clear differences in the foxtail millet yield and *WUE* under the combined application patterns of the five fertilizers (Table 3). The yield of $N_{90}P_{45}$, compared with $N_{45}$, increased by 54.43%, while the yields of $N_{60}P_{40}$–O and $N_{60}P_{40}$–M increased by 19.31% and 23.17% in 2017. However, in 2018, the yields of $N_{60}P_{30}$, $N_{90}P_{45}$, $N_{60}P_{40}$–O, and $N_{60}P_{40}$–M increased by 59.86%, 54.26%, 24.11%, and 13.12%, respectively, compared with $N_{45}$. During 2017–2018, the *WUE*s of $N_{60}P_{40}$–O and $N_{60}P_{40}$–M increased by 12.89–29.20%. Significantly, these findings revealed that the yield and *WUE* could be increased by reducing the inorganic fertilizer input and by increasing the application of organic manure and microbial manure.

**Table 3.** Effects of different manure application patterns on millet yield and *WUE*.

| Treatment | Water Consumption/(mm) | | Yield/($10^3$ kg·hm$^{-2}$) | | *WUE*/(kg·hm$^{-2}$·mm$^{-1}$) | |
| --- | --- | --- | --- | --- | --- | --- |
| | **2017** | **2018** | **2017** | **2018** | **2017** | **2018** |
| $N_{45}$ | 544.3 | 492.2 | 2.6 ± 0.2 b | 2.8 ± 0.6 b | 4.8 ± 0.4 b | 5.7 ± 1.1 b |
| $N_{60}P_{30}$ | 537.4 | 491.9 | 3.4 ± 0.9 ab | 4.5 ± 0.4 a | 6.4 ± 1.7 ab | 9.2 ± 0.8 a |
| $N_{90}P_{45}$ | 520..2 | 492.9 | 4.0 ± 0.6 a | 4.4 ± 0.9 a | 7.7 ± 1.1 a | 8.8 ± 1.7 a |
| $N_{60}P_{40}$–O | 538.2 | 492.7 | 3.1 ± 0.5 ab | 3.5 ± 0.4 ab | 5.7 ± 0.9 b | 7.1 ± 0.9 ab |
| $N_{60}P_{40}$–M | 518.7 | 492.4 | 3.2 ± 0.8 ab | 3.2 ± 1.0 ab | 6.2 ± 1.5 ab | 6.5 ± 2.1 ab |

Note: Different lowercase letters in the same column indicate significant differences at the α = 0.05 level.

## 4. Discussion

The root system is at the center of the physiological and metabolic activities of most crops [31] and plays an important role in plant biomass accumulation, as well as efficient nutrient and water absorption and utilization. It is also closely related to the yield of crops [32]. Studies have shown that different fertilization patterns affect root morphological growth and further affect crop yield. On the one hand, organic manure not only contains abundant nutrients that can improve soil structure but also has a slow-release effect after fertilization, which can ensure that the demand for mineral nutrients is met in various stages of crop growth [33]. It can also significantly promote root growth and development, deep soil root distribution, and crop root vitality [34], thus improving quality and increasing production. Microbial manure contains nitrogen-fixing bacteria. The reasonable and correct application of microbial fertilizer can reduce the loss of nutrients in organic nitrogen fertilizer and improve the appearance and quality of crops [35], while the application of manure substitution chemical fertilizer, as an organic fertilizer, increases inorganic nutrition availability in soil [36]. On the other hand, the crop yield can be improved by combining organic and inorganic fertilizers, thus supporting the sustainable development of modern agriculture [13]. It is not known which manure application modes can act as substitutes for chemical fertilizer to promote root development and increase yield; more detailed, long-term experiments are required to investigate this question [37,38]. In this study, the root parameters for $N_{60}P_{40}$–O increased by 61.59%, 16.78%, 38.78%, 33.40%, and 35.21% in the case of the *RL*, *RSA*, *RV*, *RD*, and *RB*. $N_{60}P_{40}$–M also had an increased *RL*, *RV*, and *RB* in the maturity stage (Figures 3–6). All of this shows that the combination of organic and inorganic fertilizers not only enables the use of a diminished amount of fertilizer but also ensures a good soil structure for crop root growth, as reflected by the better results obtained using organic and microbial fertilizer application modes.

Root vitality reflects the strength of root metabolism and is one of the significant indicators used to measure root function [6,39]. During the pustulation stage, the root system developed. The total length of the root system was significantly positively correlated with the average diameter, and the root vitality was improved. The fine roots increased and became active, and mostly coarse roots of the foxtail millet were observed during the mature stage. Conversely, the correlation between the total length of the foxtail millet root system and the average diameter was weak, and the root vitality decreased in terms of physiological function (Figure 8), thus supporting the conclusion of a previous study showing that a combination of organic and inorganic fertilizers is beneficial for the development of cotton root morphology and the improvement of root vitality [40].

Water is a fundamental necessity for human well-being and ecosystem sustainability; however, its toxicity due to agrochemical usage for food production leads to the deterioration of water quality. This poor water quality diminishes water reusability, thus limiting efficient water usage. [41]. Davies reported that the moderate application of nitrogen fertilizer is beneficial to increase maize yield and *WUE*, but high nitrogen fertilization leads to a lower soil nutrient utilization efficiency and prevents maize yield increase [42]. Research has also shown that the rational application of organic and inorganic fertilizers can increase soil organic matter content, improve soil moisture status, and thus improve *WUE* [13].

Some studies have also shown that the combination of organic and inorganic fertilizers can significantly improve soil physical and chemical properties, as well as increase crop yield and *WUE* [43]. These findings indicate that in agricultural production in arid and semi-arid areas, the combination of organic and inorganic fertilizers is of great significance for water storage and conservation, as well as for the improvement of *WUE* [44]. The results of our experiment indicate that, under the long-term application of nitrogen fertilizer, the *WUE* and yield significantly increased over two years. Although the *WUE* and yield increased with the application of nitrogen–phosphorus fertilizer combined with organic matter and microbial manure, the values were lower than those for inorganic fertilizer. This may be due to the significant impact of nitrogen–phosphorus fertilizer on crops and its lesser impact on soil water retention capacity. The combination of organic and inorganic fertilizer can increase soil water retention capacity, thereby promoting a stable crop yield (Table 3). By applying nitrogen fertilizer alone, the yield of foxtail millet did not reach its optimal level for two years (Table 3). Increasing the application of nitrogen and phosphorus fertilizers increased the yield of foxtail millet, with continued application resulting in an increased yield in 2017 and a decrease in 2018 (Table 3). This further demonstrates that rational fertilization can improve foxtail millet yield and *WUE*. Adebayo [45] also reported that the root system architecture traits of a crop were clearly affected by increasing or decreasing N application rates and plant densities. Under treatment with medium- and high-nitrogen fertilizers, it was easy to obtain a higher yield of maize, but excessive nitrogen fertilizer inhibited the increase in yield [14]. In this study, nitrogen and phosphorus fertilizer with organic manure or microbial fertilizer could increase foxtail millet yield with greater stability than nitrogen and phosphorus fertilizer alone. However, we did not study the effects of fertilization on the growth and development of foxtail millet roots at different depths (Table 3).

In summary, the development of root morphology has a particularly important correlation with the increase in yield. The reasonable application of organic and inorganic fertilizers promotes the root growth of foxtail millet, thereby increasing its yield. However, there were differences in the root development among the fertilization strategy explored in this experiment and those employed in research by other scholars. This may be due to the insufficient duration of our experiment, which did not achieve the best effect in improving the yield of foxtail millet in the northern Shaanxi region. Therefore, effect of the combined application of organic and inorganic fertilizers on foxtail millet yield increase in northern Shaanxi will be studied further in our subsequent experiments.

## 5. Conclusions

In the Loess Plateau, the combined application of organic manure with inorganic fertilizer can improve the effectiveness of soil nutrients, regulating the morphological development of foxtail millet roots and promoting the high resource use efficiency of the root system, although this is also affected by annual rainfall. Organic manure or microbial fertilizer combination with NP fertilizer significantly enhanced the formation of foxtail millet grains by improving root morphological construction, root elongation, and root activity, a finding supported by the fact that the foxtail millet yields increased by 54.43% and 59.86% and the *WUE*s increased by 62.39% and 59.93% in the case of $N_{90}P_{45}$ and $N_{60}P_{30}$, respectively. In addition, the combination of chemical fertilizer with organic manure or microbial fertilizer stably improved the foxtail millet yield by 13.12–24.11%, largely resulting from the strengthened correlation between root development and shoot biomass accumulation. In conclusion, the combined application of chemical fertilizer with organic manure or microbial fertilizer has clearer effects on the soil environment, root growth regulation, and yield increase than that of a combined NP input or nitrogen fertilizer used alone in the semi-arid regions of China.

**Author Contributions:** Conceptualization, T.Z., H.Z. and Q.L. methodology, L.W. and X.W.; software, T.Z.; validation, T.Z., L.W., H.Z., Q.L. and X.W.; formal analysis, X.W.; resources and data curation, T.Z., H.Z. and X.W.; writing—original draft preparation, T.Z.; writing—review and editing, T.Z.; visualization, T.Z.; supervision, H.Z.; project administration, X.W. All authors have read and agreed to the published version of the manuscript.

**Funding:** This work was supported by the National Natural Science Foundation of China (41967013, 31751001), the Natural Science Foundation of Shaanxi Province (2021JZ-55), and the Yulin Science and Technology Bureau research Project (CXY-2022-70, CXY-2022-68).

**Institutional Review Board Statement:** Not applicable.

**Informed Consent Statement:** Not applicable.

**Data Availability Statement:** Data for this research can be found at the following data link (https://pan.baidu.com/s/1AfTzkMPbw1RlkYJJga7tlA?pwd=jdxf, accessed on 12 October 2023).

**Conflicts of Interest:** The authors declare no conflict of interest.

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
