# Peer review of "Effects of Fertilizer Application Patterns on Foxtail Millet Root Morphological Construction and Yield Formation during the Reproductive Stage in the Loess Plateau of China"

_agronomy, doi:10.3390/agronomy13112847_

Round 1
Reviewer 1 Report
Comments and Suggestions for Authors
Zhao et al analyzed the effect of various fertilizer compositions, and organic and microbial manure on millet production. The study is interesting, which shows a detailed analysis of root morphology which may have influenced the water use efficiency and yield. As the results are interesting, the introduction and discussion have many gaps and the opportunity to improve it which should be addressed before publication. Some suggestions are mentioned below:
1- The introduction did not describe the role of root morphology and how it is influenced by fertilizer, organic, or microbial manure. These are described in the discussion which I think is not the correct way as the readers won't be able to relate how the introduction is connected to the results where lots of analysis are done on roots.
2- The authors highlighted water use efficiency (WUE) which gets improved in organic and microbial manure treatment but is less than the NP fertilizer. Here, I would encourage authors to cite many papers that show organic and microbial manure increases water water-retaining capacity of the soil. It is known that NP fertilizer has a bigger role in vegetative growth and less on plant defenses (e.g., plant metabolites) and the water-retaining capacity of the soil which is why it is important to use more organic and microbial manure. Some citations are cited below but would encourage authors to look for many papers that are recently published. https://doi.org/10.1007/s11356-021-15258-7; https://doi.org/10.1038/s41477-020-0656-9.
3- please mention what the difference is between organic and microbial manure. Organic manure also can contain naturally occurring microbes. It is possible here. It needs to be discussed also.
Comments on the Quality of English Language
I would request for a professional assistance with the English. It gets difficult to understand sometimes. lines 44 to 46. lines 192 to 193. These are some examples to mention but many more are there.
Author Response
Dear editor/reviewer,
Glad to receive your suggestions for modification. We wrote this email to explain the revision of the manuscript after all authors discussion. We have incorporated these changes in the revision with fonts representing revisions. We like to thank you for the important contributions to our manuscript.
Please find our detailed responses below to all these comments/suggestions. Thank you again for everything you have contributed, and I look forward to your approval.
1.The introduction did not describe the role of root morphology and how it is influenced by fertilizer, organic, or microbial manure. These are described in the discussion which I think is not the correct way as the readers won't be able to relate how the introduction is connected to the results were lots of analysis are done on roots.
Response: Thank you for this question, we revised the introduction added root morphology and references “Root morphological traits and spatial distribution play an important role in the growth and development of plants use. Root system morphological traits, include root length, root surface area, volume and density at various depths, and root length in different diameter classes. Nitrogen fertilizer can significantly improve the root activity, can improve root length, root surface root biomass and root volume of the foxtail millet and an appropriate amount of nitrogen fertilizer can improve the root activity, but excessive nitrogen application will inhibit the root activity. Wei et al. have shown that the combination of organic and inorganic fertilizers can significantly improve root length, root surface area, and volume, enhance rice root vitality, enhance nutrient absorption and utilization by roots, and provide sufficient nutrients for plant growth, which is beneficial for the accumulation of dry matter in various parts of the plant and the increase of rice yield” on line 69-80. And introduce how organic and microbial manure affect root morphology “Soil microbial play a crucial role in maintaining soil fertility at appropriate levels and improving soil structure by affecting the aggregation of soil particles, and microbial fertilizer could improve the relationship between plants roots and soil water, provide drought resistance protection in part, make plants less susceptible to certain soil borne diseases simultaneously, which means that include diseases caused by fungi that produce additional fungal toxins and reduce the occurrence of pests probably. Microbial fertilizer applied to seeds or farmland, with aims at increasing soil fertility and plant growth by increasing the diversity and biological activity of microorganisms in the rhizosphere. soil is a complex system influenced by multiple factors, improving the beneficial microbial community in soil is an important way in the biogeochemical cycling of inorganic and organic nutrients, especially in the rhizosphere, where microorganisms activity improve the availability of plant nutrients, soil quality, and the crop root growth” on line 48-60. It can also significantly promote root growth and development, promote deep soil root distribution, and improve crop root vitality. Thus, achieving the effect of improving quality and increasing production. Microbial manure contains nitrogen fixing bacteria. Reasonable and correct application of microbial fertilizer can reduce the loss of nutrients in organic nitrogen fertilizer and improve the appearance quality of crops” on line 388-393.
2.The authors highlighted water use efficiency (WUE) which gets improved in organic and microbial manure treatment but is less than the NP fertilizer. Here, I would encourage authors to cite many papers that show organic and microbial manure increases water water-retaining capacity of the soil. It is known that NP fertilizer has a bigger role in vegetative growth and less on plant defenses (e.g., plant metabolites) and the water-retaining capacity of the soil which is why it is important to use more organic and microbial manure. Some citations are cited below but would encourage authors to look for many papers that are recently published.
Response: Thank you for your question, we have added the importance of WUE to the discussion on line 416-419. Further analyzed the impact of organic and inorganic fertilizer combination on WUE, “Research has also shown that the rational application of organic and inorganic fertilizers can increase soil organic matter content, improve soil moisture status, and thus improve WUE. Some studies have also found that the combination of organic and inorganic fertilizers can significantly improve soil physical and chemical properties, increase crop yield and WUE. It indicates that in agricultural production in arid and semi-arid areas, the combination of organic and inorganic fertilizers is of great significance for water storage and conservation, as well as improving WUE. The results of this experiment indicate that under the condition of long-term application of nitrogen fertilizer, WUE and yield significantly increased over two years. Although WUE and yield of nitrogen phosphorus fertilizer combined with organic matter and microbial manure increased, they were lower than those of inorganic fertilizer. This may be due to the significant impact of nitrogen phosphorus fertilizer on crops and the smaller impact on soil water retention capacity. The combination of organic and inorganic fertilizer can increase soil water retention capacity, thereby promoting stable crop yield.”and inserted some recently published citations on line 424-438.
3.Please mention what the difference is between organic and microbial manure. Organic manure also can contain naturally occurring microbes. It is possible here. It needs to be discussed also.
Response: Thank you for this question, we discussed the difference between organic manure and microbial fertilizer. Although organic manure may also contain some microbial fertilizers, but the main function was improving the soil structure by a higher organic content; the microbial fertilizer enhanced the effectiveness of soil available nutrients by microbial activity and diversity, and their functional subjects are different. We added this part on line 385-393.
4.I would request for a professional assistance with the English. It gets difficult to understand sometimes. lines 44 to 46. lines 192 to 193. These are some examples to mention but many more are there.
Response: Thank you for this valuable comment, we have optimized English through helped from a professional assistance (Ying-long,Chen, The University of Western Australia, yinglong.chen@uwa.edu.au)with the English.
"Agronomy-2685905-Manuscripts revised after round -1 review. word" We deeply appreciate your consideration of our manuscript, and we look forward to receiving comments from you. If you have any questions, please do not hesitate to contact me at the address below.
Thank you and best regards.
Yours sincerely
Corresponding author: Xiao-Lin Wang
College of Life Sciences, Yulin University, 719000, Yulin, China
Tel: +86-15129978707
E-mail: wangxl8304@163.com

Reviewer 2 Report
Comments and Suggestions for Authors
attached

Author Response
Dear editor/reviewer,
Glad to receive your suggestions for modification. We wrote this email to explain the revision of the manuscript after all authors discussion. We have incorporated these changes in the revision with fonts representing revisions. We like to thank you for the important contributions to our manuscript.
Please find our detailed responses below to all these comments/suggestions. Thank you again for everything you have contributed, and I look forward to your approval.
1.Abstract: The abstract provides a concise summary of the key aspects of the study, including the objectives, methods, main findings, and conclusions. Overall, the abstract is well-written. However, the abbreviations of treatments should be identified before using them as they may confuse the reader.
Response: Thank you for this question, we have determined the abbreviation of treatments for the processing of the abstract, “which composted of chemical N, P, organic manure and microbial manure showed as that the single application of 45 kg·ha-1 nitrogen fertilizer as control (N45), combined application of 60 kg·ha-1 nitrogen fertilizer with 30 kg·ha-1 phosphorus fertilizer (N60P30), combined application of 90 kg·ha-1 nitrogen fertilizer with 45 kg·ha-1 phosphorus fertilizer (N90P45), 60 kg·ha-1 nitrogen fertilizer and 40 kg·ha-1 phosphorus fertilizer with 2000 kg·ha-1 organic manure (N60P40-O), and 60 kg·ha-1 nitrogen fertilizer with 40 kg·ha-1 phosphorus fertilizer with 5 kg·ha-1 microbial manure (N60P40-M). And modified it on line 17-22.
- For introduction--Lines 72-77: The introduction could provide more details on the crop traits and environmental limitations
Response: Thank you for your suggestion, we have added some details on the foxtail millets’ traits and environmental limitations (As unexpected high temperature, heavy rainfall, productive level, backward tillage technology and the use of chemical fertilizers) which affect foxtail millet grain yield and WUE, on lines 98-113.
- In the section of “Materials and methods”, there are twelve points that need to be modified or explained, and we will explain them separately below.
Line 120: more than 12%, please be precise.
Response: Thank you for your question, we were rechecked up the P2O5 content in phosphate fertilizer on line 159, and finalize 12%.
4.125: what do you mean by 20 billion/g, please describe the microbial manure. Is it solid?
Response: Thank you for this question, we wrote 20 billion·g-1 mean microbial manure content efficacious living-cell 20 billion·g-1. Microbial manure is wettable powder composed of a variety of high-activity and content probiotics and their metabolites that can fix nitrogen, solubilize phosphate and release potassium.
5.Line 126: what do you mean by solve potassium?
Response: Thank you for your question, we want to express microbial manure have an effect on release potassium and it has the ability to decompose soil organic matter, release of ineffective phosphorus in soil, balancing soil acidity and alkalinity.
6.Line 126: Please justify why did you use bacteria that fix nitrogen, dissolve phosphorus and potassium while you added soluble inorganic fertilizer of NPK. I think the mineral fertilizer will defiantly inhabit the bacterial effects.
Response: Thank you for your suggestion. The field experiment was located at the intersection of the Loess Plateau and the Mu Us Sandy Land. The soil content organic matter was lower (nearly 0.3%), the soil content availability nutrients was lower, the mircroal function was weak. The experiment used organic manure, mircobial manure substitute for partial replacement of chemical fertilizer, though added organic matter, mircroal diversity to activating soil intrinsic nutrient availability. Achieving the combination of land use and maintenance. Finally, the simultaneous improvement of organic matter content, microbial diversity, and crop yield.
7.128: please justify the selected application rates of different fertilizers you used in the experimental work. Why did you use 5 kg/ha of microbial manure?
Response: Thank you for this valuable comment, because microbial manure is wettable powder, we used the best applied amount as 5 kg·ha-1 of microbial manure introduced by manufacturer and pursued a significant effect on the soil micro-environment finally.
8.Lines 129-133: The methods section on experimental design and crop management lacks some details and could be expanded.
Response: Thank you for this suggestion, we were added some details “The planting density of foxtail millet was 180000 plants·ha-1. Seedlings were established based on their density. The fertilizer application time for each treatment were the same and timely irrigation of crops due to water shortage, 2 times of intertillage, artificial weeding and other management measures were the same as the requirements for field experiments.” on the methods section on experimental design and crop management on line 180-184.
- Line: 139-144: please re-write this section again to be more understandable. “Finally, the foxtail millet is harvested uniformly when it is fertilizer”… what do you mean?
Response: Thank you for your question, we have re-write this section on line 169-173. The last sentence of this paragraph we want to express the foxtail millet is harvested uniformly when it was mature completely.
10.Line 144: kg ha-2, please re-visit
Response: Thank you for this suggestion, we were re-visit 144 kg·ha-1 and modify it on the line 193.
11.Lines 151-153: More information is needed on the root washing method and storage conditions before analysis.
Response: Thank you for this valuable comment, we have added some methods (used a nylon mesh sieve to remove most of the soil, then wash under a slow flow water and lean the residual soil with a brush.) on the root washing and storage conditions before analysis on line 201-203.
12.Line 127-133: The subsection on experimental design and management could be organized more clearly into separate paragraphs on field preparation, treatment application, crop management, data collection.
Response: Thank you for this suggestion, we have modified it, moved its location made them separate into paragraphs and optimized its content on line 169-192.
13.Line 151-162: Consider organizing the root data collection and analysis methods into separate paragraphs for clarity
Response: Thank you for this suggestion, we would divide 2.4.1 into two paragraphs on line 197-214. The first paragraph we introduced how digging and washing root, and the second paragraph we wrote methods of analyzing root and measured some indexes.
14.Lines 163-170: The yield and yield component data collection requires more details on sampling and measurements.
Response: Thank you for your suggestion, we were added more details into the yield component (20 same plant-samples were randomly collected in the middle of each plot using soil drilling method collected profile root at maturity, after harvested, the spike lengths, spike diameter, weight per spike, grain weight per spike, 1000-grain weight and yields were measured in laboratory. The Spike length (cm) measured from the top of the spike to the base using a tape and spike diameter (mm) of the foxtail millet measured with Vernier caliper with each measurement at the center of the spike, respectively. Cut at the base of the foxtail millet spike, and weighted it with an electronic balance with an accuracy of 0.001, then recorded data, respectively. And then all mature spikes were harvested.) data collection make this experiment more clarify and it was added on line 216-226..
15.Lines 257-263: The results for root biomass effects are unclear and could be presented more effectively.
Response: Thank you for your suggestion and question, we were reanalysis the results for root biomass added into some results and re-draw the image on line 320-329.
16.Lines 364-371: The discussion could be expanded to interpret the differences in root responses between fertilizer treatments.
Response: Thank you for your suggestion, we were optimized the discussion, “In this study, the root parameters in N60P40-O can significantly increase by 61.59%, 16.78%, 38.78%, 33.40%, 35.21% in RL, RSA, RV, RD, RB, N60P40-M has also increased RL, RV, RB, at maturity stage (Fig 3-6), all of these showed that the combination use of organic and inorganic fertilizers not only diminished fertilizer quantity, but also ensured a good soil structure for crop root growth, the better results carried out by using organic and microbial fertilizer application modes in the study.” expanded to interpret the differences in root responses between fertilizer treatments, and added some experimental results on line 399-405.
17.Lines 372-385: The limitations of the study duration and scope should be acknowledged.
Response: Thank you for this question, we have added the limitations of this study (on the Loess Plateau) and at the end of the conclusion, the scope of application of this experiment was written (in the semi-arid region of China) on line 465-468.
18.Lines 386-391: The conclusion highlights yield benefits but does not summarize key findings on root growth effects.
Response: Thank you for your suggestion, we were summarize key findings on root growth effects “Organic manure or microbial fertilizer combination with NP fertilizer significantly enhanced foxtail millet grains formation by improving root morphological construction, root elongation and root activity, which supported by that foxtail millet yields and WUEs increased by 54.43% and 59.86%, 62.39% and 59.93% in N90P45 and N60P30.” added into the conclusion on line 468-472.
"Agronomy-2685905-Manuscripts revised after round -1 review.word"We deeply appreciate your consideration of our manuscript, and we look forward to receiving comments from other reviewers. If you have any questions, please do not hesitate to contact me at the address below. And I would like to obtain some suggestions from you.
Thank you and best regards.
Yours sincerely
Corresponding author: Xiao-Lin Wang
College of Life Sciences, Yulin University, 719000, Yulin, China
Tel: +86-15129978707
E-mail: wangxl8304@163.com

Round 2
Reviewer 1 Report
Comments and Suggestions for Authors
The authors have taken the time to improve the quality of the paper. No further reservations are required.
Author Response
thanks for your review and suggestion, our paper has a great promotion after revised following your valuable comments, thanks again.